# Drug Delivery Systems: Study of Inclusion Complex Formation between Methylxanthines and Cyclodextrins and Their Thermodynamic and Transport Properties

**DOI:** 10.3390/biom9050196

**Published:** 2019-05-20

**Authors:** Cecília I. A. V. Santos, Ana C. F. Ribeiro, Miguel A. Esteso

**Affiliations:** 1Department of Chemistry, Coimbra University Centre, University of Coimbra, 3004-535 Coimbra, Portugal; anacfrib@ci.uc.pt; 2U.D. Química Física, Universidad de Alcalá, 28871 Alcalá de Henares (Madrid), Spain; miguel.esteso@uah.es

**Keywords:** drug delivery systems, xanthines, cyclodextrin, multicomponent diffusion, apparent molar volumes, viscosity

## Abstract

This paper presents an analysis of the molecular mechanisms involved in the formation of inclusion complexes together with some structural interpretation of drug–carrier molecule interactions in aqueous multicomponent systems comprising methylxanthines and cyclodextrins. The determination of apparent partial molar volumes ($\varphi_{V}$) from experimental density measurements, both for binary and ternary aqueous solutions of cyclodextrins and methylxanthines, was performed at low concentration range to be consistent with their therapeutic uses in the drug-releasing field. The estimation of the equilibrium constant for inclusion complexes of 1:1 stoichiometry was done through the mathematical modelling of this apparent molar property. The examination of the volume changes offered information about the driving forces for the insertion of the xanthine into the cyclodextrin molecule. The analysis on the volumes of transfer, Δ$\varphi_{V,c}$, and the viscosity *B*-coefficients of transfer, Δ*B*, for the xanthine from water to the different aqueous solutions of cyclodextrin allowed evaluating the possible interactions between aqueous solutes and/or solute–solvent interactions occurring in the solution. Mutual diffusion coefficients for binary, and ternary mixtures composed by xanthine, cyclodextrin, and water were measured with the Taylor dispersion technique. The behavior diffusion of these multicomponent systems and the coupled flows occurring in the solution were analyzed in order to understand the probable interactions between cyclodextrin–xanthine by estimating their association constants and leading to clearer insight of these systems structure. The measurements were performed at the standard (298.15 ± 0.01) K and physiological (310.15 ± 0.01) K temperatures.

## 1. Introduction

Since the origins of our civilization, humans have tried to improve their health and relieve or eliminate pain through the ingestion or administration of certain substances. Empirical knowledge coupled with industrial development has allowed the advance of medical sciences, and specifically in the pharmaceutical area, of new therapeutic agents from both natural and synthetic origins [1,2,3]. However, not all these substances possess the necessary properties to be effective therapeutic agents. They often do not go beyond the first phase of clinical trials, which is, in the opinion of most researchers [4,5], due to poor pharmacokinetic properties in humans, lack of clinical efficiency, high adverse effects, or even due to economic factors [6,7]. One way to evade this situation is to create alternatives to conventional forms of administration, e.g., through controlled drug delivery systems [8,9].

A controlled (or modified) drug release system [10] employs a carrier molecule within which the drug is incorporated, being later released at a certain rate in a defined time and place. The main goal is simple: to minimize the problems arising from conventional administration forms, since the effectiveness of a drug in vivo is largely conditioned by its physicochemical characteristics, particularly by its solubility in biological fluids and on cell membranes [11]. With a controlled drug release system it may be possible to achieve a reduction in adverse effects together with an extended time of activity (maintaining the drug concentration in the plasma constant) or even protection against enzymatic attacks or degradation effect of pH.

Cyclodextrins are one of the most commonly used groups of carriers to encapsulate drug molecules of limited water solubility, enhancing their stability and improving bioavailability [12,13,14,15,16,17,18]. They are cyclic oligosaccharides, comprising a variable number of D-(+)-glucose molecules, linked together by α-(1,4)-type bonds. Native cyclodextrins (Figure 1) are constituted by 6, 7, or 8 glucose units and are referred to as α-, β-, γ-cyclodextrins, respectively [19,20]. Its structure is in the form of truncated cone, where primary hydroxyl groups are located in the narrower end of the cone and the secondary hydroxyl groups located on the opposite side, at the wider end [21].

Xanthines are possibly the most widely used pharmacological agents at a global level [22] and are naturally present in everyday consumer products such as coffee, tea, and chocolate. Structurally, the methylxanthines caffeine and theophylline (Figure 2) are purinergic derivatives formed by condensed pyrimidine and imidazole rings [23]. They act as psychostimulants, i.e., on the central nervous system (SNS), increasing motor activity and intellectual performance, reducing fatigue and sleep. Their actions as drugs are similar (due to its chemical analogy) and their specificity is related to the location of the methyl groups in the molecule.

Caffeine and theophylline are weak bases with *K*_a_ values of 0.78 mol kg^−1^ [24] and 0.94 [25], respectively, and water solubility of 2.17 g/100 mL and 0.55–0.8 g/100 mL (25 °C) [26], respectively; both are soluble in lipids. The pharmacological action of caffeine is characterized by blocking adenosine receptors in the brain [27,28], with the effects produced being dependent on the dose. Theophylline presents the ability to relax smooth muscle (bronchi and pulmonary blood vessels) and to have a stimulatory effect on breathing. It also stimulates the myocardium and central nervous system, decreases peripheral pressure and venous pressure, and has a diuretic effect [28,29]. Theophylline bronchodilator activity is essentially due to the (nonspecific) inhibition of adenosine and phosphodiesterase receptors [30].

Based on the above considerations, these drugs are good models for developing drug-controlled release systems in a drug–CD complex model:(a)they have limited applicability due to solubility limits;(b)the administration of methylxanthines has high adverse effects at concentrations above the therapeutic range, which is especially important in the case of theophylline, due to its small therapeutic range, which requires the dose and the form of administration to be rigorously controlled until the effective concentrations are reached.

This paper aims to explore the characteristics of drug delivery systems composed by cyclodextrins and methylxanthines in order to overcome the main disadvantages of the administration of the individual drugs. Although the usual way to study cyclodextrin–drug complexes is through solubility measurements, this chapter’s goal is to present an alternative method: volumetric, viscosimetric, and diffusion determinations performed to qualitatively and quantitatively estimate the existence of interactions between these molecules in aqueous solution, as well as the dimensions of the existing free and associated species and association constants, consequently attaining information on the type and properties of complexes or aggregates that can form [31].

## 2. Mathematical Concepts

### 2.1. Molar Volume

The apparent molar volume of a solute in an aqueous solution with molality *m* can be calculated from the equation
(1)$$\varphi_{V}=\frac{V-V_{H_{2}O}}{m}=\frac{M}{\rho }+\frac{1000}{m}\;\left(\frac{1}{\rho }-\frac{1}{\rho_{H_{2}O}}\right)$$
at where ρ and $\rho_{H_{2}O}$ are, respectively, the density of the solution and the pure water at the working temperature; *M* is the molar mass of the solute; *V* is the volume (in cm^3^) of the solution with molarity *m* and *V_H2O_* is the volume (in cm^3^) of pure water.

The extrapolation of the values of $\varphi_{V}$ toward infinitesimal concentration can be attained by using the Masson equation [32], in the case of electrolytes,
(2)$$\varphi_{V}=\varphi_{V}^{0}+S_{V}^{0}\sqrt{c}$$
and according to Redlich [33] in the case of non-electrolytes,
(3)$$\varphi_{V}=\varphi_{V}^{0}+b_{V}c$$
where $S_{V}^{0}$ and $b_{V}$ are the slope of the corresponding fitting which can be related to the interactions between the solutes in the solution.

Terekhova et al. [34,35] proposed that in a mixture of solutes in solution, and considering Young’s rule for a interaction model 1:1 between two associating molecules, e.g., between one molecule of cyclodextrin and one molecule of xanthine in solution, the volume $\varphi_{V}$ can be expressed in terms of the volumes of all species present in the system, free and associated, according to
(4)$$\varphi_{V}=\left(1-\alpha_{c}\right)\varphi_{V,f}+\alpha_{c}\varphi_{V,c}$$
where α_c_ is the fraction of associated molecules and $\varphi_{V,f}$ and $\varphi_{V,c}$ are the volumes of free and fully associated species, respectively. In the case of a cyclodextrin plus a drug, we can write:(5)$$\alpha_{c}=\frac{Km_{CD,f}m_{Drug,f}}{m_{CD}}$$
where
(6)$$m_{CD}=m_{CD,f}\left(1+Km_{Drug,f}\right)$$
*m_CD_* is the stoichiometric concentration of the cyclodextrin, and *m_CD,f_* and *m_Drug,f_* are the concentrations of the cyclodextrin and the drug in the free state, respectively. The manipulation of Equations (4) to (6) would allow the expression
(7)$$\varphi_{V,CD}=\frac{\varphi_{V,f}+K\varphi_{V,c}m_{Drug,f}}{1+Km_{Drug,f}}$$

Thus, by determining the apparent molar volume of the solutes, carrying out measurements under conditions where each component of the solution is in the presence of constant concentration of the other component, the association constant *K* can be estimated.

### 2.2. Viscosity

Viscosity is the physical property of the fluids which characterizes the resistance they oppose to movement, due to an external force. The examination of this transport property allows an overview of the interactions occurring in the solution at the solute/solvent level.

For electrolyte solutions, Jones and Dole [36] developed an equation that allows concerning the relative viscosity of an electrolyte solution to the concentration of the solution
(8)$$\frac{\eta }{\eta_{0}}=1+Ac^{1/2}+Bc+Dc^{2}$$
where *η* is the viscosity of the solution and *η_0_* is the viscosity of the solvent, and *A*, *B*, and *D* are temperature-dependent parameters. Coefficient *A* provides information about long-range intermolecular forces (solute–solute interactions) and its value can be used as an indicator of the occurrence of some type of association. Coefficient *B* is related to solute–solvent interactions that take place in the solution and can provide information about the organizing (*structure-making*) and disorganizing (*structure-breaking*) nature of the solvent structure. That is, a positive *B* coefficient is associated with solutes with an organizing capacity of the structure of the solvent (*structure-making*). Thus, for example, if the solvent is water, this capacity indicates that the solute is strongly hydrated in the solution and, therefore, the change in the viscosity of the solution is accentuated with the increase in concentration. Likewise, its presence in the medium can promote the formation of hydrophobic aggregates in the aqueous solution. On the contrary, a negative *B* coefficient is associated with solutes that have a breaking capacity of the structure of water (*structure breaking*) and of hydrophobic aggregates in aqueous solution and, in addition, indicates that the solute is weakly hydrated. This classification correlates quite well with others raised by other authors, as is the case of the Hofmeister series (which classifies the ions as cosmotropic or chaotropic and orders them according to their capacity to stabilize proteins and membranes). Coefficient *D* is related both to the solute–solute and solute–solvent interactions and is important only for high electrolyte concentrations and will not be considered in this analysis.

For non-electrolyte solutions, the study of the influence of concentration on the viscosity can be accomplished on the basis of the Einstein equation [31,37],
(9)$$\frac{\eta }{\eta_{0}}=1+Bc+Dc^{2}$$
being *B* and *D* constants depending on both the nature of the solute and the solvent, as well as temperature and pressure. *B* coefficient can be assumed to be equivalent to that of the Jones–Dole equation (Equation (8)), so that it will also reflect the effects of solute–solvent interactions on the solution. The sign and magnitude of the *D* coefficient, in turn, can be associated with the solute–solute interactions that occur in the medium [31].

### 2.3. Mutual Isothermal Diffusion

Diffusion is an important irreversible process taking place as a result of a gradient of concentration inside the solution [38,39]. Its quantitative measure can be done through the diffusion coefficient of each species in solution [40].

In a multicomponent system, Fick’s first law [38] relates, for a given component *i* and confined to a one-dimensional space, the flow of matter *J_i_* with its concentration gradient through its diffusion coefficient, *D*, according to:(10)$$-\left(J_{i}\right)=\sum_{j=1}^{q}D_{ij}\frac{\partial c_{j}}{\partial x}\left(i=1,2,\ldots ,q\right)$$
where *c* and ∂c/∂x represent the concentration and its gradient in the *x* direction, respectively; *D*_ij_ represents the effect of the flux of component *j* on the behavior of component *i*; and *q* is the number of independent components in the system. From Equation (10), the diffusion coefficients can be experimentally determined.

For a ternary system (*q* = 3; two solutes plus a solvent) [39], in a constant volume referential ν and taking into account that it is possible to eliminate the terms of the solvent flow since it is interrelated with those of the solutes, Equation (10) for solutes 1 and 2 is expressed as:(11)$$-(J_{1})={\left(D_{11}\right)}_{v}\frac{\partial c_{1}}{\partial x}+{\left(D_{12}\right)}_{v}\frac{\partial c_{2}}{\partial x}$$
(12)$$-(J_{2})={\left(D_{21}\right)}_{v}\frac{\partial c_{1}}{\partial x}+{\left(D_{22}\right)}_{v}\frac{\partial c_{2}}{\partial x}$$
where *J*_1_ and *J*_2_ are the molar fluxes of components 1 and 2; ∂*c*_1_/∂*x* and ∂*c*_2_/∂*x* are the concentration gradients of solutes 1 and 2; *D*_11_ and *D*_22_ are the main diffusion coefficients, and represent the flow of solute generated by its own chemical potential gradient; and *D*_ik_ are the cross diffusion coefficients representing the coupled flux of solute *i* produced by the concentration gradient of the solute *k*. If a cross diffusion coefficient, *D*_ik_ (*i* ≠ *k*), is positive, this means that coupled co-transport of solute *i* takes place from the higher to the lower concentration regions of solute *k*, which reinforces the total flux of the *i* solute. On the contrary, if this *D*_ik_ (*i* ≠ *k*) is negative, this means that such co-transport occurs in opposite sense, i.e., solute *i* moves from the lower to the higher concentration regions of solute *k*, which reduces the total flux of the *i* solute.

For multicomponent diffusion in which several solutes are involved, the information on the possible intermolecular interactions is very important for the understanding of the diffusion process. In that sense, models have been developed to quantify these molecular interactions by means of estimating an association constant. One of these models has been developed by Paduano et al. [41,42,43,44] considering the particular case of two solutes in chemical equilibrium, where species 1 is a cyclodextrin (1), species 2 is a host molecule (2), and species 3 is the possible complex, and defined the diffusion equations in terms of the present species as
(13)$$-(J_{1}{}^{*})={\left(D_{11}{}^{*}\right)}_{\nu }\frac{\partial c_{1}*}{\partial x}+{\left(D_{12}{}^{*}\right)}_{\nu }\frac{\partial c_{2}*}{\partial x}+{\left(D_{13}{}^{{}^{*}}\right)}_{\nu }\frac{\partial c_{3}*}{\partial x}$$
(14)$$-(J_{2}{}^{*})={\left(D_{12}{}^{*}\right)}_{\nu }\frac{\partial c_{1}*}{\partial x}+{\left(D_{22}{}^{*}\right)}_{\nu }\frac{\partial c_{2}*}{\partial x}+{\left(D_{23}{}^{{}^{*}}\right)}_{\nu }\frac{\partial c_{3}*}{\partial x}$$
(15)$$-(J_{3}{}^{*})={\left(D_{31}{}^{*}\right)}_{\nu }\frac{\partial c_{1}*}{\partial x}+{\left(D_{32}{}^{*}\right)}_{\nu }\frac{\partial c_{2}*}{\partial x}+{\left(D_{33}{}^{{}^{*}}\right)}_{\nu }\frac{\partial c_{3}*}{\partial x}$$
where the secondary terms *D*_ij_^*^ give the interactions between solutes produced solely due to the diffusion process. In terms of the constituent species, we can consider the following relations
(16)$$J_{1}=J_{1}{}^{*}+J_{3}{}^{*}\;\mathrm{and}\;J_{2}=J_{2}{}^{*}+J_{3}{}^{*}$$
(17)$$c_{1}=c_{1}{}^{*}+c_{3}{}^{*}\;\mathrm{and}\;c_{2}=c_{2}{}^{*}+c_{3}{}^{*}$$
and the association constant in molarity is
(18)$$K_{c}=\frac{c_{3}{}^{*}}{c_{1}{}^{*}c_{2}{}^{*}}$$

Neglecting all the secondary terms in Equations (13) to (15) and rearranging the equations, the association constant, *K*_c_, and the diffusion coefficient of the associated species, *D*_33_^*^, can be obtained by
(19)$$D_{11}=\frac{1}{2}\left\{\;(D_{11}{}^{*}+D_{33}{}^{*})+(D_{11}{}^{*}-D_{33}{}^{*})\left[1\;-\;K_{c}(c_{2}-c_{1})\right]R\right\}$$
(20)$$D_{12}=\frac{1}{2}\left\{\;(D_{33}{}^{*}-D_{11}{}^{*})+(D_{11}{}^{*}-D_{33}{}^{*})\left[1-K_{c}(c_{2}-c_{1})\right]R\right\}$$
(21)$$D_{21}=\frac{1}{2}\left\{\;(D_{33}{}^{*}-D_{22}{}^{*})+(D_{22}{}^{*}-D_{33}{}^{*})\left[1-K_{c}(c_{2}-c_{1})\right]R\right\}$$
(22)$$D_{22}=\frac{1}{2}\left\{\;(D_{22}{}^{*}+D_{33}{}^{*})+(D_{22}{}^{*}-D_{33}{}^{*})\left[1-K_{c}(c_{2}-c_{1})\right]R\right\}$$
where
(23)R={[1+Kc(c2−c1)]2+4Kcc1}−12

The model described implies that the estimation of *K*_c_ from Equations (19) to (22) is based on the assumption that *D*_11_^*^ and *D*_22_^*^ are equal to the binary diffusion coefficients, *D*_1_ (cyclodextrin) and *D*_2_ (guest molecule) at the corresponding concentration, corrected for the solution viscosity.

## 3. Experimental

### 3.1. Materials and Solutions

Caffeine (*pro analysi*, purity > 98.5%), theophylline (anhydrous, purity > 99.0%), *β*-cyclodextrin (98% purity, 13.1% water mass fraction), 2-hydroxypropyl-*β*-cyclodextrin (100% purity, 18.2% water mass fraction), and potassium chloride (*pro analysi*, purity > 99.5%) were provided by Sigma-Aldrich and used as received, having only been kept in a desiccator over silica gel. Milli Q water (κ = 5.6 × 10^−8^ S cm^−1^) was used as solvent. Solutions were prepared by directly weighing both the solute and the solvent, with an accuracy of ±0.0001 g. Cyclodextrin water content was considered when calculating the solution concentration. Molality (*m*, mol kg^−1^) of the solutions contains an uncertainty of less than 0.07% [31].

### 3.2. Equipment and Procedure

Density values were carried out with an Anton Paar DMA5000M densimeter (accuracy of 5 × 10^−6^ g cm^−3^ in the temperature range of 0–90 °C and the pressure range of 0–10 bars). The Peltier system that this equipment incorporates guarantees a temperature variation inside the vibrating U-tube better than ±0.005 degrees. Measurements were accomplished at 298.15 K and 310.15 K. They were performed at least in quadruplicate, taking the average value (when providing a reproducibility better than 0.001%) as the value of the solution density [31].

Viscosity values (with a reproducibility of ±0.001 mPa s) were obtained by using an Ostwald-type viscometer, immersing in a water thermostat bath with temperature variation of ±0.02 degrees. The efflux time (obtained as the average value of, at least, four runs) was measured by using a digital stopwatch with a resolution of 0.2 s. These times were always higher than 300 s, so the kinetic energy correction (Hagenbach correction) was not applied [31].

Diffusion coefficient measurements were performed by using the Taylor dispersion method, which is profusely described in the literature [39,45,46,47,48,49,50,51], and whose scheme is presented in Figure 3. In a summarized way, we will point out that a small amount of solution under study (samples of 0.063 mL at concentration (*c*_j_ ± Δ*c*)) is injected into a carrier solution of concentration *c*_j_ (point 4) that moves in laminar flow throughout a long capillary tube (of 3.2799 (±0.0001) × 10^4^ mm in length and 0.5570 (±0.00003) mm of radius).

The outlet of this dispersion capillary tube is connected to the differential refractometer which compares the actual refraction index value to that of the carrier solution, translating the difference into an electric voltage signal depending on the elapsed time, *V*(t). These voltages can be split into:(24)$$V\left(t\right)=V_{0}+V_{1}t+V_{max}{\left(t_{R}/t\right)}^{1/2}\exp[-12D{\left(t\;–\;t_{R}\right)}^{2}/r^{2}t]$$
*V*_0_, *V*_1_, and *V*_max_ are the baseline voltage, the baseline slope, and the peak height, respectively; and *t*_R_ is the mean sample retention time.

In case of multicomponent systems (ternary or quaternary systems), for which high order mutual diffusion coefficients (*D*_ik_) are involved, it is necessary to replicate two or more pairs of peaks for each run. In such a case, the fitting equation becomes into:(25)$$V(t)=V_{0}+V_{1}t+K\sum_{i=1}^{n}R_{i}\left[c_{i}(t)-\bar{c_{i}}\right]$$
where *V* is the detected voltage signal; *K* = d*_V_*/d*n* is the sensitivity of the detector, *n* is the refractive index; *R_i_* = d*n*/d$\bar{c_{i}}$ measures the change in the detected property per unit change in the concentration of the solute, and $c_{i}(t)-\bar{c_{i}}$ represents the dispersion solute average concentration [31,46,52,53].

## 4. Results and Discussion

### 4.1. Characterization of Physicochemical Properties of a Drug in Aqueous Solution

As indicated above, the efficacy of a drug is largely conditioned by the value of its solubility in water and in the present of the other components. Therefore, knowledge of the physicochemical properties of the aqueous solutions of these compounds is important.

For each of the binary systems under investigation, both drug + water and cyclodextrin + water systems, density, viscosity, and diffusion coefficients in aqueous solution were obtained and analyzed at different concentrations and at temperatures of 298.15 and 310.15 K.

#### 4.1.1. Caffeine + Water System

Apparent molar volumes, (Appendix A) [54], obtained from density measurements, for caffeine in water were related to the concentration by using the Masson equation (Equation (2)) at both studied temperatures. It was observed that they decrease with increasing solute concentration in aqueous solution, indicating an associative interaction between caffeine molecules, as it was already reported by other authors [24,55]. Falk et al. [56] studied caffeine through FT-IR measurements and suggested that the hydrogen bonds between the C=O groups of caffeine and water would decrease as association between solute molecules occurs, releasing water into the medium. The entrance of caffeine into aqueous solution and the corresponding formation of its hydration sphere would lead to the establishment of caffeine–water hydrogen bonds that would generate a contraction of the solution volume [57,58]. At the same time, these hydrated solute entities could enter the cavities of the ice-like structure of water by hydrophobic hydration [59,60] (decreasing the total free energy of the solution). The increase of these entities in solution would lead to a smoothing of this effect as a consequence of the decrease of the number of free interstices in the water structure [61,62]. From the Masson equation, the values of the apparent molar volumes at infinitesimal concentration were also obtained, (142.5 cm^3^ mol^−1^ and 146.5 cm^3^ mol^−1^ at 298.15 K and 310.15 K, respectively) and found to be in good agreement with data previously published by other authors [55,63]. The increase in $\varphi_{V}^{0}$, as a consequence of the increase in temperature, confirms the previous hypothesis that caffeine has a *structure-making* character in aqueous solution.

From the fitting of the viscosity values of aqueous solutions of caffeine (Appendix A) [54] to the Jones–Dole equation, additional information can be obtained. Small and negative *A* values were observed at both temperatures, which allows us to infer the presence of caffeine–caffeine interactions, especially at 298.15 K [55,56,64]. Positive values for the *B* coefficient were found as well as for its derivative against temperature, d*B*/dT (= 0.06), which confirms the anomalous *structure-making* character of this solute. On the other hand, high values of $B/\bar{V_{2}^{0}}$ (above 2.5) indicate the existence of an ordered first layer of hydration around the caffeine [65,66], which is consistent with the statement that this is a *structure-making* solute.

Concerning the diffusion coefficients for this system (Appendix A) [54], their behavior as a function of temperature and concentration was studied. Diffusion coefficient values at infinitesimal concentration of caffeine were obtained from the fitting of *D* against concentration (*D*^0^ = 0.764 × 10^−9^ m^2^ s^−1^ at 298.15 K and *D*^0^ = 1.077 × 10^−9^ m^2^ s^−1^ at 310.15 K) and also through experimental measurements by injecting samples of low concentration (c < 10 mM) into laminar flows of water (*D*^0^ = 0.760 × 10^−9^ m^2^ s^−1^ at 298.15 K and *D*^0^ = 1.052 × 10^−9^ m^2^ s^−1^ at 310.15 K) [31,54]. Both sets of values agree very well and are within the imprecision that accompanies this experimental method.

From the Stokes–Einstein equation, values of *r* = 0.321 nm and *r* = 0.305 nm were estimated for the hydrodynamic radius with a corresponding molar hydrodynamic volume of 83.47 cm^3^ mol^−1^ and 71.61 cm ^3^ mol ^−1^, at 298.15 and 310.15 K, respectively, showing that at higher temperature, the caffeine molecule tends to be less hydrated.

#### 4.1.2. Theophylline + Water System

Appendix A summarizes the apparent molar volume values after fitting to Masson equation. Although they are smaller than those of caffeine, as it would be expected due to the absence of the methyl group at position 7, they also show a decrease with concentration, which can be attributed to self-association. This behavior also suggests that theophylline must have a *structure-making* character.

The fitting parameters of the viscosity experimental data to the Jones–Dole equation are presented in Appendix A. As it can be observed, *A* parameter shows higher negative values than those found for caffeine, indicating a distinct self-associative capacity with interactions between the theophylline molecules stronger than those produced between caffeine molecules, especially at 310.15 K. The *B* coefficient value is positive, and the values for (d*B*/dT = 0.06) and $B/\bar{V_{2}^{0}}$ are equal to those of caffeine at both temperatures, which means the *structure-making* character of theophylline in aqueous solution, reinforcing the previous conclusion from the analysis of the apparent molar volumes. A value of *B* for theophylline lower than that for caffeine is in agreement with the fact that the theophylline molecule has one less methyl group; therefore, it is more polar (less hydrophobic) than caffeine and it is likely that the structure of water is less affected by the self-association of this solute [67,68].

Appendix A shows the diffusion coefficients’ values. They decrease with increasing theophylline concentration in solution, with this decrease being more pronounced in the case of the higher temperature. Estimated limiting diffusion coefficients (*D*^0^ = 0.811 × 10^−9^ m^2^ s^−1^ at 298.15 K and *D*^0^ = 1.152 × 10^−9^ m^2^ s ^−1^ at 310.15 K) are very close to those experimentally determined (*D*^0^ = 0.806 × 10 ^−9^ m^2^ s^−1^ and *D*^0^ = 1.157 × 10^−9^ m^2^ s^−1^, at (298.15 and 310.15) K) by injections of dilute theophylline solutions into a laminar flow of water. By applying the Stokes–Einstein equation, values of *r* = 0.302 nm (for 298.15 K) and *r* = 0.285 nm (for 310.15 K) were obtained for the hydrodynamic radius. As with caffeine, the rise in temperature seems to produce a dehydration of the theophylline molecule. The estimated values for the molar hydrodynamic volume were 71.08 cm^3^ mol^−1^ at 298.15 K and 57.90 cm^3^ mol^−1^ at 310.15 K, as expected, lower than those found for caffeine [31,54].

#### 4.1.3. *β*-Cyclodextrin (*β*-CD) + Water System

Apparent molar volumes, calculated from density values, are positive and high (Appendix A) [55]. The value at 298.15 K is nearly concentration independent, in concordance with published data [69,70,71,72]. On the contrary, at 310.15 K, a descendent trend with the concentration is observed, with no literature data available for comparison. Apparent molar volumes present large values, consistent with those commonly found in organic compounds of high molecular mass.

As expected, the influence of the concentration on the viscosity of these *β*-CD solutions showed an increase with the concentration, which was more pronounced at the physiological temperature (Appendix A) [54]. *D* value has to be disregarded, due to its noteworthy low value at both temperatures together with a high uncertainty. The values found for the *B* coefficient are large and positive (indicating the formation of a hydration layer around the cyclodextrin molecule in aqueous solution), which, in accordance to Marcus [73], indicates that *β*-cyclodextrin has a *structure-making* capacity, in agreement with the previous analysis of the volumetric results as well as with the behavior reported for other cyclodextrins [52]. However, from the value (d*B*/dT = 0.10) this solute would have a *structure-breaking* character [74]. Despite this apparent contradiction, it is necessary to remember that theories about the organizing or disorganizing character of the solvent structure are directed to electrolytes, and this solute is a non-electrolyte. Marcus [73] and other experts agree that this characterization is primarily based on the sign and magnitude of *B*, so this is the consideration followed.

The diffusion coefficients (Appendix A) [69,75] decreases with increasing *β*-CD concentration in the medium, but only about 2%, possibly due to its large size and, consequently, the need for that molecule to find a suitable contiguous space to be able to move within the solution [76]. The experimental and estimated limiting diffusion coefficient, at 298.15 K, *D*^0^ = 0.326 × 10^−9^ m^2^ s^−1^, is coincident with that obtained by other authors [69]. At 310.15 K, the observed value was *D*^0^ = 0.464 × 10^−9^ m^2^ s^−1^. The estimated hydrodynamic radius was *r* = 0.752 nm at 298.15 K and *r* = 0.747 nm at 310.15 K, values very close to those reported by Longsworth [77] for this solute and those of Evans et al. [78] for similar solutes. The variations in molar hydrodynamic radii with temperature are less than 1%, so this cyclodextrin should not be affected by the increase in temperature. The hydrodynamic volumes obtained were 1074.3 cm^3^ mol^−1^ and 1021.9 cm^3^ mol^−1^ at 298.15 K and 310.15 K, respectively.

#### 4.1.4. Hydroxypropyl-*β*-Cyclodextrin (HP-*β*-CD) + Water System

Apparent molar volumes, calculated from the density measurements (Appendix A), are considerably higher than those of *β*-CD, possibly due to the substitution of the hydroxyl groups by larger hydroxypropyl groups, and tend to increase with the concentration of the solution. Apparent molar volume at the infinitesimal concentration obtained by using Redlich equation (Equation (3)) [32] were (861 and 880.76) cm^3^ mol^−1^ at (298.15 and 310.15) K.

The analysis of the viscosity of the HP-*β*-CD solutions (Appendix A) using the Einstein equation (Equation (9)), resulted in almost negligible values for the *D* coefficient (Appendix A), whereas *B* values were large and positive, and higher than 2.5, at both temperatures, showing that the solute is solvated in solution and therefore has a *structure-making* capacity. However, as occurred for the *β*-CD, the variation of *B* with temperature, d*B*/dT presents a positive value which, according to Nightingale [70], indicates a disorganizing character of the solvent structure, supported by the positive variation of the apparent molar volume, with the increase in concentration previously observed. If we take into account that HP-*β*-CD has exterior hydroxypropyl groups in substitution for the *β*-CD hydroxyl groups, increasing its solubility and the hydrophilic character, the conclusion is that this cyclodextrin has greater capacity to establish bonding with the surrounding water molecules, and that for this it would be necessary to break the structure of the solvent.

Concerning the diffusion coefficients (Appendix A) [79], values slightly decrease with the increase in cyclodextrin concentration, mainly at 298.15 K. The increase of viscosity in the medium caused by a greater presence of solute is compensated by a decrease of the structure of the solvent. This situation translates into a diminution of *D* less pronounced than expected due to the viscous effect, especially at the higher temperature, where the decrease produced in the structure of the solvent, either by thermal effect or by a greater hydration of the solute, practically compensates the viscosity increase of the medium. Diffusion coefficient values obtained from the fitting against concentration were *D*^0^ = 0.322 × 10^−9^ m^2^ s^−1^ at 298.15 K and *D*^0^ = 0.408 × 10^−9^ m^2^ s^−1^ at 310.15 K, which are close to the results previously obtained for *β*-CD. The values obtained for the hydrodynamic radius of HP-*β*-CD were estimated as *r* = 0.76 nm and *r* = 0.80 nm for (298.15 and 310.15) K, respectively, and for the molar hydrodynamic volumes were (1115 and 1317) cm^3^ mol^−1^ at (298.15 and 310.15) K, respectively. These values are larger than those for *β*-CD, which confirms a greater hydration of the HP-*β*-CD.

### 4.2. Influence of Cyclodextrins on the Physicochemical Properties of Methylxanthines in Aqueous Solution

Cyclodextrins in aqueous solution can form inclusion complexes in which the water molecules of the cavity are replaced by host molecules (wholly or partially included) with a hydrophobic character. However, the outer hydroxyl groups of the cyclodextrin molecule may bond with other molecules or even with contiguous cyclodextrin molecules. Loftsson et al. [80,81] proposed that in saturated aqueous solution, cyclodextrin inclusion and non-inclusion drug complexes may coexist, and this could explain why the equilibrium constant value is sometimes concentration-dependent and reliant on the experimental method applied in its determination. Despite the intense research in this field, there is an area for which few studies have been directed: the modulation of the properties of the host–guest interaction at physiological temperature. Furthermore, it is also important to understand the effect of the temperature on the association of the drugs with different cyclodextrins. In order to analyze the type of interactions occurring in these solutions, thermodynamic and transport properties for ternary systems (drug + cyclodextrin + water) were measured and used for the estimation of both the association constants and the size of the species in solution.

#### 4.2.1. Characterization of the Interactions between Cyclodextrins and Caffeine in Aqueous Solution

Assuming the association between a cyclodextrin molecule and a caffeine molecule in aqueous solution giving rise to a 1:1 complex, which may or may not be of inclusion, and that the association process is given by
(26)$$CD+Caf\;\overset{K}{↔}\;CD\cdot Caf$$
where *CD* represents the cyclodextrin molecule, *Caf* represents the caffeine molecule, and *K* is the corresponding equilibrium constant, the variations in the values of the latter will be a consequence of the changes occurring in the medium, due to the association of the solutes. Thus, the analysis of the changes in the physicochemical properties of the solution as a consequence of the interaction between the solutes can work as an alternative way of quantifying the equilibrium constant.

In ternary mixtures, i.e., cyclodextrin + guest + water, the apparent molar volume measured for each component can be expressed in terms of the volumes of all species present in solution, free and associated. According to Terekhova et al., it would be possible in this way to obtain the fraction of each associated species from apparent molar volumes and relate it to the molality of the solution (Equations (4) to (7)), in order to obtain the equilibrium constant of this association.

From the density measurements for the aqueous solutions of caffeine in the presence of each of the cyclodextrins (Appendix A) [54], the limiting values for the apparent partial molar volumes were estimated and compared with those for the caffeine in pure water, thus obtaining the transfer volumes from water to a mixed solvent (Appendix A) [54], were calculated in accordance with $\Delta V_{\phi }^{0}=\left[V_{\phi }^{0}\left(\mathrm{water}-\beta -\mathrm{CD}\;\mathrm{solution}\right)-V_{\phi }^{0}\left(\mathrm{water}\right)\right]$.

At the infinitesimal concentration, the interactions between the solute molecules fade and the observed transfer volumes are solely the result of the interactions between the solute and solvent molecules. If we consider the water + cyclodextrin mixture to be a mixed solvent, then the information provided by the values of the partial molar transfer volumes will be relative to the type of interaction between the caffeine and cyclodextrin molecules [31,54]. Values of $\Delta \varphi_{V}^{0}$ are predominantly positive at low concentrations of cyclodextrin, and decrease, becoming negative, with increasing concentration of the cyclodextrins, at both temperatures. For caffeine in the presence of HP-*β*-CD, at 298.15 K, the transfer volumes show opposite behavior.

Due to their hydrophobic cavity, the cyclodextrin molecules provide the caffeine molecule with a suitable environment for interaction to occur and inclusion complexes to form. The compatible exterior of the cyclodextrins with water allows hydrogen bonding, avoiding the caffeine self-association and leading to the dominance of interactions between the two solutes and consequently a decrease in the net volume. As temperature increases, free spaces are produced in the ordered solvent medium, resulting in a better fit of the structured complexes, decreasing the predominantly hydrophobic contribution of the caffeine–cyclodextrin interaction, and increasing the partial molar volume.

The volume changes that a solute undergoes when being transferred from water to a mixed solvent can be explained by cosphere overlap model developed by Friedman and Krishnan [82], according to which the overlapping effect of hydration shells is destructive. The superposition of the hydration shells of two ionic species results in an increase in volume, while the superposition of the hydrophobic spheres of two hydrophobic groups or of a hydrophobic group and an ionic group contributes negatively to the volume.

An analysis of the type of interactions that may occur between caffeine and cyclodextrin molecules in aqueous solution allows the classification into different groups [83]:

1—hydrophilic–ionic interactions between OH groups of cyclodextrin molecules and charged points of the caffeine molecule (C=O or N);

2—hydrophilic–hydrophilic type interactions among OH groups of the cyclodextrin molecules and the C=O group of the caffeine molecule, facilitated through hydrogen bonds;

3—hydrophilic–hydrophobic type interactions concerning the OH groups of cyclodextrin molecules and non-polar groups of the caffeine molecule (-CH_3_);

4—hydrophobic–hydrophobic interactions between non-polar cyclodextrin molecules groups (-CH_2_-) and caffeine molecule (-CH_3_).

The superposition of the ion hydration cosphere (> CO- and > N-) and the hydrophilic OH group (type 1 interaction) makes a positive contribution to the volume, resulting in the decrease of the *structure-breaking* tendency of the ion and a reduction of the electrolysis of water caused by these ions. The same result comes from the overlap of the hydration cosphere of OH groups (type 2 interaction), heading to an increase in the magnitude of the hydrogen bonding interaction. In contrast, the interactions of type 3 and 4 cause a shrinkage in the transfer volume.

The trend observed in Appendix A [54], with volumes increasingly more negative suggest that, in ternary solutions, hydrophobic–hydrophobic or hydrophobic-type interactions predominate over hydrophilic–ionic or hydrophilic groups. The only exception is for the case of caffeine in the presence of HP-*β*-CD, at 298.15 K, where there seems to be a predominance of type 1 and 2 interactions, seeing that with increasing concentration of this cyclodextrin the volumes become more and more positive.

Indeed, it is assumed that the more significant contribution to the thermodynamics of the complexation process of cyclodextrins [84] arises mainly from the penetration of the hydrophobic part of the host molecule into the cyclodextrin cavity and, to a lesser extent, from the dehydration of the guest. The occurrence of hydrogen bonding, when possible, may function as a third factor with a stabilizing effect on the complex. The release of water molecules and the conformational changes subsequent to the entrance of the host molecule may also contribute to the complexation. All these types of interactions influence the molar volume of the compounds in solution, and since the changes deriving therefrom can be quantified; therefore, the values for the association constants of these molecular systems can be obtained. The analytical resolution of Equation (7) by the least squares method allows obtaining $\varphi_{V}$ (apparent molar volume) for each of the components, as well as the *K* (stability constant) and $\varphi_{V,c}$, the apparent molar volume of the complexed species (Appendix A) [54].

A detailed inspection of Appendix A shows that the association of both cyclodextrins with caffeine is characterized by stability constants whose values are close to literature at 298.15 K [85,86,87,88,89,90,91] for studies performed in aqueous solution. Still, for caffeine in the presence of HP-*β*-CD, the results are comparable even to those under different pH conditions. Taking into account that biological fluids present different pHs in different places of the organism (for example, in the stomach pH ≈ 2 and in the blood pH ≈ 7.4), it is interesting to note that, in principle, this factor does not significantly influence the stability constant and therefore the results are valid as approximation to the general physiological conditions. The increase in temperature leads, as expected, to a decrease in the stability constant of caffeine–cyclodextrin association. For caffeine in the presence of HP-*β*-CD at 310.15 K, there are no changes in volume that gave indication that association was occurring, or it was insufficient to enable quantification. Generally, the stronger the stability of the complex, the smaller the volume variation associated with its formation. The caffeine molecule undergoes volume changes, resulting from the association process, that are higher in the presence of *β*-CD than in the presence of HP-*β*-CD. These changes may be due to the partial substitution of the hydroxyl groups by hydroxypropyl groups (in HP-*β*-CD) around the macrocyclic cavity of the cyclodextrin, thus altering the affinity for the caffeine molecule and indicating that the nature of the caffeine–cyclodextrin association is not solely due to the interactions with the interior of the cavity, as previously perceived through the analysis of partial molar transfer volumes.

Applying the same model for the analysis of the caffeine volume variation in the case of complex formation, the values of $\Delta \varphi_{V,c}$ for the caffeine-*β*-CD system can be interpreted taking into account the reorganization of the solvent molecules when the complex is formed, leading to a positive volume variation due to the replacement of water molecules by a caffeine molecule [92,93,94]. Additionally, the formation of the complex is accompanied by a partial destruction of the hydration cospheres of the two solutes, followed by its restructuration, due predominantly to the overlapping of hydrophobic groups, and producing a decrease in volume [82] which can be further enlarged by the formation of hydrogen bonds [95]. The sign of $\Delta \varphi_{V,c}$ results from the predominance of one contribution over another, in this case clearly hydrophobic. When caffeine is in the presence of HP-*β*-CD, the volume changes are smaller and positive, indicating a predominance of hydrophilic type 1 and type 2 interactions. Aicart et al. [89] performed molecular mechanics simulations for the inclusion complexes for this system through the Hiperchem v.5.1 program, and obtained geometries that show that the structural unit of the caffeine purine ring remains at the entrance of the cyclodextrin cavity, leaving the methyl groups and C=O exposed, the latter being responsible for the establishment of hydrophilic interactions that result in an increase in volume.

Examination on the viscosities of aqueous solutions of caffeine in the presence of cyclodextrin may also reveal what kind of interactions occur in solution, mainly through the analysis of the Jones–Dole *B*-coefficient providing information on both the solubility of the solutes and their influence on the solvent structure around the molecules of the solute. The viscosities of aqueous solutions of caffeine in the presence of *β*-CD and HP-*β*-CD were measured at 298.15 K and 310.15 K, and the Δ*B* coefficients of transfer from water to different aqueous solutions of cyclodextrin were calculated (Appendix A) [54]. The Jones–Dole *B*-coefficient values for caffeine are positive and rise with increasing concentration of cyclodextrin in solution, leading to the interpretation that there are solid solute–solvent–cosolvent interactions which are more important, the higher the cyclodextrin content in the solution. Assuming that the *B*-coefficient provides information regarding the interactions that occur between solute and solvent molecules (in this case, a mixed solvent), resulting in a larger or smaller structuring of the solution [96], then in these solutions, caffeine behaves as a *structure-making* electrolyte at both temperatures. In addition, as mentioned earlier, the greater the amount of cyclodextrin, the more structured are the solutions; i.e., the greater the number of bonds between the molecules.

Transfer Δ*B* coefficients for caffeine from water to the mixed solvent confirm the results discussed above (based on apparent molar volumes) that the existence of cyclodextrin molecules in solution gives caffeine the suitable conditions for the formation of complexes (which may or may not be of inclusion), and that the interaction between caffeine and cyclodextrin predominates over the self-association of the caffeine molecules.

Taking into consideration that diffusion is one of the major ways of movement of substances between cells and one of the essential modes drugs cross the cell membrane, it is important to comprehend how affected this process can be by the existence of cyclodextrins in the environment and what effect the temperature has on it. The interest in the study of the diffusion properties of ternary systems encompassing a cyclodextrin molecule and a guest molecule (in aqueous solution) is related to the estimation of the equilibrium constant for the inclusion process, under reasonable assumptions, from the four experimentally determined diffusion coefficients [41,42,43,44].

The measurements of the mutual diffusion coefficients of aqueous solutions of caffeine in presence of *β*-CD and HP-*β*-CD [97,98,99] (Appendix A), were carried out using the Taylor dispersion technique, allowing us to obtain the diffusion coefficients that describe these systems *D*_11_, *D*_12_, *D*_21_, and *D*_22_ at temperatures of 298.15 K and 310.15 K.

The main diffusion coefficients, *D*_11_ and *D*_22_, provide the molar fluxes of the cyclodextrin (1) and caffeine (2) components caused by their own concentration gradient. The secondary diffusion coefficients, *D*_12_ and *D*_21_, provide the coupled molar fluxes of each solute induced by the concentration gradient of the other solute. Although the latter should be zero at infinitesimal concentration, for solutes interacting as it is the case, and at finite concentrations, their nonzero values may provide information on the effect of these macromolecular solutes on the diffusion of caffeine.

For the case of the *β*-CD (1) + caffeine system (2) (Appendix A) [97,99], the main diffusion coefficients are lower than their corresponding binary values measured by the same experimental technique (deviations of less than 3% at both temperatures). The addition of caffeine has little effect on *D*_11_ for *β*-CD, but the addition of the latter has considerable effect on the *D*_22_ of caffeine. The association of caffeine molecules with cyclodextrin in solution, through the eventual formation of inclusion complexes, would result in a lower mobility of the caffeine molecules and consequently a decrease in *D*_22_. The fact that this effect is less pronounced in *D*_11_ than in *D*_22_ may be due to the similarity between the free cyclodextrin mobilities and the *β*-CD-caffeine aggregates. The secondary diffusion coefficients can be used to estimate the coupled transport of solutes using the *D*_21_/*D*_11_ and *D*_12_/*D*_22_ ratios, being observed that a mole of diffusive *β*-CD continuously co-transports about 0.05 moles of caffeine and that this value increases up to 0.13 mol as the temperature increases, and that a mole of diffusing caffeine carries a maximum of 0.01 mol of *β*-CD at 298.15 K. At 310.15 K, the *D*_12_ values are small and close to the experimental error.

For the HP-*β*-CD (1) + caffeine (2) system (Appendix A) [98,99], we can see a dissimilar effect due to the presence of this cyclodextrin. At 298.15 K there is a decrease in the main diffusion coefficients between 5 and 8% with respect to the corresponding binary systems. At 310.15 K, the *D*_11_ coefficient for HP-*β*-CD does not appear to be influenced by the presence of other species. In contrast, the *D*_22_ values are substantially smaller than its correspondent binary diffusion coefficients (deviances from 3 to 15%) and analog ternary diffusion coefficients *β*-CD. For a given concentration of HP-*β*-CD, the diminution is more marked with increasing caffeine. Here again we can find evidence of the presence of caffeine and HP-*β*-CD complexes in solution, with or without inclusion, being its interaction more ample than for *β*-CD. In addition, *D*_12_ and *D*_21_, show an increase that cannot be neglected, compared to those presented by the *β*-CD (1) + caffeine (2) system. Both solutes generate coupled fluxes of the other component in solution and, for the range of concentrations studied, a mole of diffusive HP-*β*-CD co-transports up to 0.2 moles of caffeine, value which increases up to 0.28 mol at 310.15 K. A mole of caffeine diffusing at 298.15 K can counter-transport approximately 0.03 mol of HP-*β*-CD at 298.15 K, or at 310.15 K a maximum of 0.09 mole of HP-*β*-CD in favor of its concentration gradient.

Diffusion data was used to quantify the interactions between solutes in solution and estimate the equilibrium constants and the diffusion coefficients of the associated (complex) species in aqueous solution that are presented in Appendix A [31].

The estimated equilibrium constants obtained by applying the model proposed by Paduano et al. [41,42,43,44] are similar both to literature and to those previously estimated through the use of volumetric measurements, although the considerations of the diffusion model do not contemplate the inclusion of the secondary terms *D*_ij_^*^, potentially giving lower estimations for the equilibrium constants.

The diffusion coefficient values of the associated species, *D*_33_^*^, are very close to those of the cyclodextrin molecule, *D*_11_^*^, for both cyclodextrins under study, which indicates that the caffeine molecule is totally or partially included in the cyclodextrin cavity (hypothesis previously advanced); therefore, the dimensions of the diffusing species, cyclodextrin or cyclodextrin–caffeine, are very similar. In the case of HP-*β*-CD-caffeine at 310.15 K, it was possible to estimate the *K* value using the diffusion measurements, although the value for *D*_33_^*^ displays some deviation from that of the free cyclodextrin, which could indicate existence of some kind of outer interactions between the cyclodextrin and caffeine molecules rather than their inclusion in the cavity [31].

#### 4.2.2. Characterization of the Interactions between Cyclodextrins and Theophylline in Aqueous Solution

There is no unanimity in the literature concerning the interaction between theophylline and the cyclodextrin molecules of our study, *β*-CD and HP-*β*-CD. In the various available studies [85,90,91,100,101], all authors agree that there is association between theophylline molecules and *β*-CD molecules, but the value of the equilibrium constant describing this association and the ratio of molecules that are associated are discordant. Ammar et al. [101] propose the formation of a complex whose stoichiometry is of the 2:1 type, while the vast majority of authors assume that the association occurs by a 1:1 mechanism. Given the analogy of structures between caffeine and theophylline, the type of occurring interactions should be similar. In the following investigation, a 1:1 ratio, assumed by most authors, was adopted.

The analysis of the apparent molar volumes at the infinitesimal concentration, $\varphi_{V}^{0}$, for theophylline in the presence of both the cyclodextrins (Appendix A) [31] as well as the partial molar transfer volumes, $\Delta \varphi_{V}^{0}$, from water to aqueous solutions of cyclodextrin, shown in Appendix A [31], revealed that the values are predominantly negative and decrease with the increase in concentration of the cyclodextrins, at both temperatures. The effect is even more pronounced for theophylline in the presence of HP-*β*-CD.

Bearing in mind that the (1–4) types of interactions described above for the caffeine–cyclodextrin system stay valid for theophylline molecule given its structural similarity, it is seen that there are increasingly negative volume variations that would indicate the predominance of hydrophobic interactions for theophylline in presence of cyclodextrin(s). Since the volume variations are practically of the same amplitude for both cyclodextrins, the inclusion may be nonspecific, i.e., not sensitive to the substitution of the hydroxyl groups by the hydroxypropyl groups. The quantification of the interactions happening in solution, applying the same model than in the case of caffeine, allowed us to obtain the values of *K* and $\varphi_{V,c}$ for these systems (Appendix A) [31]. It can be confirmed that there is association between theophylline and cyclodextrin molecules in aqueous solution. Values obtained for the association constants are distinct from those in the literature for analogous conditions, i.e., the calorimetric measurements for the *β*-CD-theophylline system indicated a stability constant of *K* ≈ 2 kg mol^−1^ and the non-occurrence of association for the HP-*β*-CD–theophylline system. Once more, it is in the presence of HP-*β*-CD that higher values for the association constant are observed, allowing us also to correct the previous hypothesis of nonspecific association. The association process generates a decrease in volume in both the associated cyclodextrin and theophylline, in general, inversely proportional to the value of the association constant.

Concerning viscosity, aqueous solutions of theophylline were studied in the presence of constant concentration of cyclodextrin (Appendix A) [31] in order to evaluate the changes occurring in this solute dissolved in a mixed solvent relative to an aqueous solution and the results were evaluated with Jones–Dole equation, the parameters *A* and *B* and the Δ*B* coefficient are shown in Appendix A. For the case of theophylline, the effects observed on the Jones–Dole *A* and *B* coefficients, due to the presence of cyclodextrin, are similar to those of caffeine. This fact indicates that the interaction between these molecules (solute–solvent–cyclodextrin) predominates over the self-association of theophylline, increasing as the amount of cyclodextrin introduced into solution increases, and translating into an increase in the solution structure, that is, in higher bonding between the molecules. When the temperature increases, these interactions are weaker.

The analysis of the data obtained for the diffusion coefficients of the cyclodextrin + theophylline systems, *D*_11_, *D*_12_, *D*_21_, and *D*_22_ systems at 298.15 K (Appendix A) [102] and 310.15 K (Appendix A) allowed us to qualify and quantify the association between these solutes. For the *β*-CD (1) + theophylline (2) system, it was found that *D*_11_ and *D*_22_ are smaller than those obtained for the corresponding binary aqueous systems, *D*_1_ and *D*_2_, with deviations between 2 and 4% for cyclodextrin and up to 18% in the case of theophylline, at the highest temperature. The secondary diffusion coefficients are generally negative and close to zero at low concentrations, within the experimental error of this method. For the higher concentrations, the secondary diffusion coefficients, *D*_12_ and *D*_21_, show a significant increase in the presence of 10 mM *β*-CD at both temperatures. Each mole of diffusing cyclodextrin can counter-transport up to 0.2 moles of theophylline, a value that decreases with increasing temperature. The cyclodextrin-coupled flux generated by the theophylline gradient is small and capable of transporting only 0.1 mole cyclodextrin per mole of diffusing theophylline, also decreasing with increasing temperature.

For the HP-*β*-CD (1) + theophylline (2) system (Appendix A) [102], the main diffusion coefficients, *D*_11_ and *D*_22_, are also lower than the binaries but show deviations up to 4% for cyclodextrin and up to 25% for theophylline, especially at the higher temperature. As for the secondary diffusion coefficients, we can see a significant increase in the *D*_21_ values, especially at the highest concentration of cyclodextrin, at both temperatures. This means that the concentration gradient of HP-*β*-CD produces coupled co-transport of theophylline, and in the compositions used, a mole of HP-*β*-CD diffusing co-carries a maximum of 0.2 moles of theophylline, and this value increases up to 0.6 moles when the temperature increases. The coupled flow of cyclodextrin generated by the theophylline gradient is small and decreases with increasing temperature.

Using the values obtained for the diffusion coefficients of these systems to quantify the interactions between solutes in solution, it was possible to obtain the equilibrium constants and the diffusion coefficients of the associated species (complexes) in aqueous solution, as presented in Appendix A [31]. The values for *K* obtained applying the model of Paduano et al. [41,42,43,44] are in good agreement with those previously estimated by volumetric measurements, again with a lower estimated *K*, but discrepant regarding the literature values. Objectively, the values found evidenced an important effect of the presence of HP-*β*-CD on theophylline in aqueous solution. If one analyzes the values found for the associated species, it could be conjectured that the association occurred at a ratio of two molecules of theophylline to each molecule of cyclodextrin, due to the marked decrease in *D*_33_^*^ compared to *D*_11_^*^, higher than expected if the molecule was totally or partially enclosed in the cyclodextrin cavity. The association could occur at both the inner and outer planes of the cyclodextrin molecule and lead to *D*_22_ and *D*_33_^*^ diffusion coefficients much lower than expected. In addition, the volume variations observed for the association process (relatively large mainly for cyclodextrin) suggests that not only the cavity would be involved in the association process, but also that there would be interactions outside the cyclodextrin involving the non-polar parts of the molecules.

## 5. Conclusions

Controlled drug delivery systems in aqueous solution, described by the cyclodextrin–drug model, were characterized through their thermodynamic (density and apparent molar volume) and transport properties (viscosity and mutual diffusion coefficients). The effect of the temperature was also studied. The study was carried out for aqueous solutions of pure drugs and pure cyclodextrin (binary systems), as well as aqueous solutions of a drug plus a cyclodextrin (ternary systems).

From the analysis of the results for the binary aqueous solutions, it can be concluded that caffeine presents a strong tendency towards self-association. On the other hand, HP-*β*-CD has a great capacity for interaction with water molecules, resulting in incorporation of a relatively large number of them into its hydration sphere.

From the analysis of the ternary systems, it is clear that both caffeine and theophylline interact with the CDs to form drug–CD complexes, where those with HP-*β*-CD, which have higher stability constants, result in the highest application of this CD at the pharmacological level. This interaction becomes weaker when the temperature increases, so the entry of the drug–CD complex into the human body can result in early release of the drug, distant from the intended site of absorption, which reduces its pharmacological efficacy.

## Figures and Tables

**Figure 1 biomolecules-09-00196-f001:**
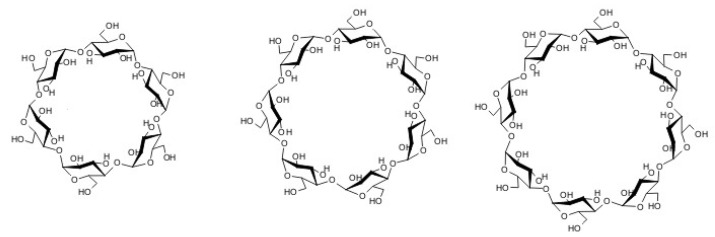
α-, β-, and γ-cyclodextrins’ structures.

**Figure 2 biomolecules-09-00196-f002:**
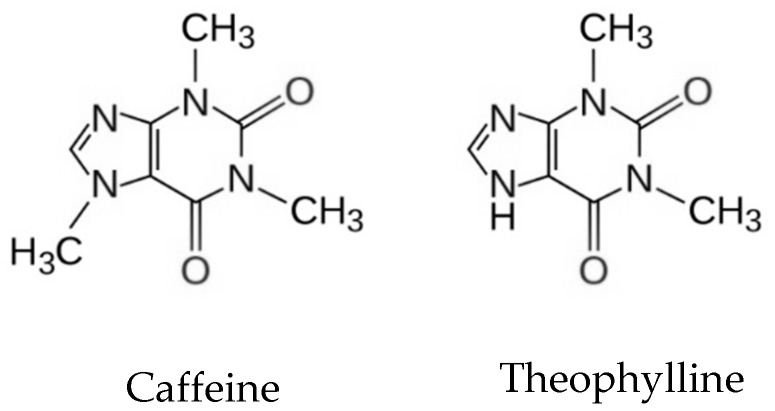
Chemical structure of caffeine and theophylline.

**Figure 3 biomolecules-09-00196-f003:**
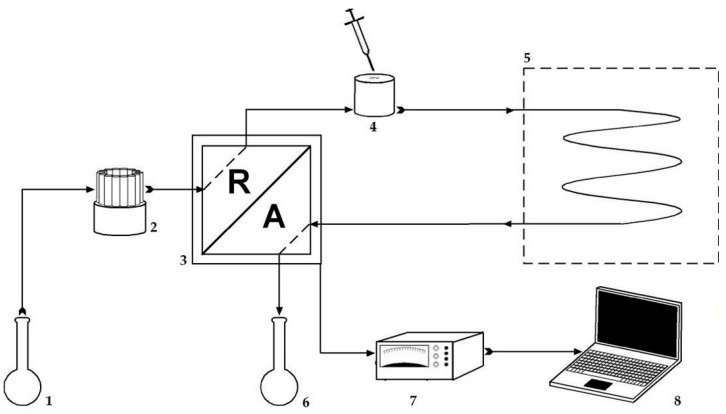
Scheme of the experimental set-up for Taylor dispersion technique. **1**—Carrier solution reservoir; **2**—Metering pump (Gilson model Minipuls 3) to keep the flow rate constant (0.17 mL min^−1^); **3**—Differential refractometer (Waters model 2410); **4**—Injection 6-port valve (Rheodyne, model 5020); **5**—Thermostatized air box ((298.15 ± 0.01) K) containing the dispersion tube; **6**—Waste material; **7**—Digital voltmeter (Agilent 34401 A; the voltage measurements were carried out at accurately 5 s intervals); **8**—Computer (connected to the voltmeter through an IEEE interface). The flow direction is indicated by the arrows.

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
