# Peer review of "Drug Delivery Systems: Study of Inclusion Complex Formation between Methylxanthines and Cyclodextrins and Their Thermodynamic and Transport Properties"

_biomolecules, 2019, doi:10.3390/biom9050196_

Reviewer 1 Report

The MS reports some interesting results on CD inclusion complexes formation investigated by “direct” thermodynamic methods. The data seems reliable and the data analysis well done on the basis of the literature reports. My main concern is that sometimes the paper is difficult to be followed specially by people that are not deeply into apparent molar properties. I would suggest a more extensive use of figures and to place the tables with the data as supporting information. Moreover, introduction should be revised reporting more recent use of CD based inclusion complexes and thermodynamic characterizations (see for instance: Journal of Thermal Analysis and Calorimetry 2015, 121 (3), 1345–1352. https://doi.org/10.1007/s10973-015-4604-2; J Therm Anal Calorim (2018) 132: 191. https://doi.org/10.1007/s10973-018-6982-8)   

Author Response

Answers to Reviewer #1 comments (modifications in the manuscript are highlighted in green)

Comments and Suggestions for Authors

The MS reports some interesting results on CD inclusion complexes formation investigated by “direct” thermodynamic methods. The data seems reliable and the data analysis well done on the basis of the literature reports. My main concern is that sometimes the paper is difficult to be followed specially by people that are not deeply into apparent molar properties. I would suggest a more extensive use of figures and to place the tables with the data as supporting information. Moreover, introduction should be revised reporting more recent use of CD based inclusion complexes and thermodynamic characterizations (see for instance: Journal of Thermal Analysis and Calorimetry 2015, 121 (3), 1345–1352. https://doi.org/10.1007/s10973-015-4604-2; J Therm Anal Calorim (2018) 132: 191. https://doi.org/10.1007/s10973-018-6982-8)   

We are grateful for the referee’s recommendation. We had transferred the tables to supporting information and revised the literature to include more recent research over CD use and their inclusion complexes thermodynamics (refs 13 to 18).

We are thankful for the referee’s suggestion to depict tables into figures. Unfortunately, many of the changes in the properties here studied are not easily detected in a visual plot. Taking into account the fundamental nature of the investigation on the properties here described, the authors consider that providing the values may be more useful  to a future user.

Reviewer 2 Report

The paper deals with the study of inclusion complex formation between caffeine and theophylline and cyclodextrins by using thermodynamic and transport properties. The manuscript is quite well organize and written. The data reported are impressive in number and of interest for quality.

The subject falls in the topics of Biomolecules and deserves for a publication providing minor revision.

More in details:

1) The authors should give a motivation on the selection of methylxantine for this study

2) The use of much more hydrophobic cyclodextrin as methylated b-CD could be of interest

3) The sentences "structure making" and "structure breaking" should be explained better

4) The terms B and D (also A) in equation 9 should be explained

5) Experimental: It is not clear if cyclodextrins were dryed up to constant weight before use

6) Did the authors considered different complex molar ratios?

Author Response

Answers to Reviewer #2 comments (modifications modifications in the manuscript are highlighted in blue)

Comments and Suggestions for Authors

The paper deals with the study of inclusion complex formation between caffeine and theophylline and cyclodextrins by using thermodynamic and transport properties. The manuscript is quite well organize and written. The data reported are impressive in number and of interest for quality.

The subject falls in the topics of Biomolecules and deserves for a publication providing minor revision.

More in details:

1) The authors should give a motivation on the selection of methylxantine for this study

We are grateful for the referee’s recommendation. Indeed in page 5, the last 2 paragraphs on introduction explain the main motivation that lead to the study of methylxantines over other pharmaceutical compounds.

2) The use of much more hydrophobic cyclodextrin as methylated b-CD could be of interest

We are grateful for the referee’s suggestion to perform experiments for a much more hydrophobic cyclodextrin as results could be interesting. Indeed the present results are only part of the extensive work done on these systems. In this work we wanted to highlight the difference that existed between the association with the most commonly excipient used – β-cyclodextrin a molecule that has hydrophobic characteristics and its replacement for a more hydrophilic molecule- HPβCD- that would result in a higher association constant for the methylxantines. This goes in line with our goal to increase the water solubility of the drug, maintaining their characteristics in solution and their bioavailability. We will take into consideration the referee’s suggestion for future work.

3) The sentences "structure making" and "structure breaking" should be explained better

We are grateful for the referee’s suggestion. Consequently, we have included that information in the text for a better comprehension.

4) The terms B and D (also A) in equation 9 should be explained

We are grateful for the referee’s suggestion and, consequently, we have included that information in the text for a better comprehension.

5) Experimental: It is not clear if cyclodextrins were dryed up to constant weight before use

We are very grateful for these comments. Regarding the referee observation about the drying of the cyclodextrin we clarify that they were not died prior to their use. However, they were kept in a desiccator over silica gel and cyclodextrin water content, and the water content, determined both based on the Sigma supplier information and by drying to constant mass at 420 K in a nitrogen atmosphere, was considered when calculating the solution concentration, as stated in point 3.1 experimental – Materials and solutions.

6) Did the authors considered different complex molar ratios?

Yes we have considered these different complex molar ratios. We have estimated the association constants considering other ratios (e.g. CD - guest 1:2 or 2:1), nevertheless, the results for the association constants and host-guest complex volumes had no scientific significance and, thus, they were not discussed in the work.

Round  2

Reviewer 1 Report

revised ms is suitable for publication. figure were not included in the revised form but data are in table format. I left the decision to editor as this is just a presentation and not a scientific issue.